# Metabolomic Study of *Dactylis glomerata* Growing on Aeolian Archipelago (Italy)

**DOI:** 10.3390/metabo12060533

**Published:** 2022-06-09

**Authors:** Manuela Mandrone, Lorenzo Marincich, Ilaria Chiocchio, Piero Zannini, Riccardo Guarino, Ferruccio Poli

**Affiliations:** 1Department of Pharmacy and Biotechnology, University of Bologna, Via Irnerio, 42, 40126 Bologna, Italy; lorenzo.marincich2@unibo.it (L.M.); ilaria.chiocchio2@unibo.it (I.C.); ferruccio.poli@unibo.it (F.P.); 2BIOME Lab, Department of Biological, Geological and Environmental Sciences, Alma Mater Studiorum—University of Bologna, Via Irnerio, 42, 40126 Bologna, Italy; piero.zannini2@unibo.it; 3Department of Biological, Chemical and Pharmaceutical Sciences and Technologies, University of Palermo, 90133 Palermo, Italy; riccardo.guarino@unipa.it

**Keywords:** *Dactylis glomerata*, Aeolian Islands, ^1^H NMR metabolomics, plant metabolites

## Abstract

The Aeolian Islands (Italy) are a volcanic archipelago in the Tyrrhenian Sea comprising seven main islands, among which are two active volcanoes. The peculiar geological features and the wide variety of environments and soils have an important impact on native plants, and in particular, the Aeolian populations of *Dactylis glomerata* (a perennial cool-season bunchgrass) exhibit remarkable phenotypic variability. Considering that environmental drivers also strongly affect the production of plant metabolites, this work aimed at comparing the metabolomic profiles of *D. glomerata* (leaves) harvested at different altitudes on four islands of the Aeolian archipelago, namely: Lipari, Vulcano, Stromboli and Panarea. Samples were analyzed by ^1^H NMR profiling, and data were treated by PCA. Samples collected on Stromboli were very different from each other and from the samples collected in the other islands. Through an Orthogonal Partial Least Squares (OPLS) model, using altitude as the *y* variable, it emerged that the concentration of proline, glycine betaine, sucrose, glucose and chlorogenic acid of *D. glomerata* growing on Stromboli decreased at increasing altitude. Conversely, increasing altitude was associated with an increment in valine, asparagine, fumaric acid and phenylalanine.

## 1. Introduction

*Dactylis glomerata* L. (Poaceae), vernacularly known as orchard grass or cocksfoot, is a perennial cool-season bunchgrass. Although in some areas it has become an invasive species, *D. glomerata* is widely grown as pasture for grazing in North America, Europe and Oceania [1,2,3,4]. Due to its shade and drought tolerance, *D. glomerata* is also gaining ecological importance for restoration of rocky landscapes undergoing desertification [4,5,6]. Moreover, because of its tendency to form dense nonrhizomatous root mat, *D. glomerata* is recommended for erosion control on forestland that has been burned or logged [7].

The allelopathic interactions between *D. glomerata* and other plants have been addressed by several studies, demonstrating the capability of *D. glomerata* to influence the growth of other plants. For instance, *D. glomerata* rhizosphere extract was able to inhibit the growth and development of companion and volunteer species [8], and aqueous extracts of *D. glomerata* inhibit seed germination and growth of red clove (*Trifolium pratense*) [9]. On the other hand, *D. glomerata* growth is influenced by other plants, such as fresh leaf litter from exotic plants (*Ailanthus altissima*, *Robinia pseudoacacia* and *Ulmus pumila*) [10], or *Calamagrostis epigejos* extracts, which inhibit the growth of *D. glomerata* roots and the germination of the seeds [11]. From *D. glomerata* leaves, different hydroxycinnamate esters, free and conjugated with flavonoids, were isolated [12,13]. Nevertheless, in the literature, there are no studies available on the metabolome of this plant.

This work focused on *D. glomerata* growing on the Aeolian Islands (Italy), a volcanic archipelago in the Tyrrhenian Sea, comprising seven main islands, among which are two active volcanoes (Stromboli and Vulcano). The peculiar geological features, including volcanic emissions, and the wide variety of environments and soils have an important impact on native plants [14,15]. In particular, *D. glomerata* of the Aeolian archipelago, when subjected to biometric and enzymatic polymorphism studies, exhibited a remarkable phenotypic variability [16]. Moreover, a highly diverse environment strongly affects plants’ metabolome, driving plants to increase and diversify the metabolite production in response to biotic and abiotic stimuli [17,18]. In order to compare a high number of plant phytochemical profiles, a metabolomic approach proved a successful strategy, which generally relies on untargeted analysis protocols, whose results are handled with chemometrics (multivariate data treatment) [19]. Metabolomics applied to plant science found a number of different applications, such as identification of active principles in medicinal plants [20,21,22], taxonomical and phylogenetic studies [23,24], food and botanical quality control and fraud detection [25,26,27], plant physiology and plant–environment interaction studies [28,29]. In particular, ^1^H NMR profiling is one of the most commonly used techniques for metabolomic studies [30], due to its robustness, high reproducibility and quick execution, which make it a very successful tool able to facilitate also the sharing of data and the building of databases. Moreover, this approach is nondestructive for the sample and provides specific information for unambiguous structure elucidation of the metabolites present in the samples. However, NMR-based metabolomics is limited by the sensitivity of the technique, which makes it inadequate to detect metabolites in trace amounts; thus, it is generally employed for a first overview of the most abundant metabolites present in the samples. 

In this work, an ^1^H NMR-based metabolomics approach was used to compare the phytochemical profiles of *D. glomerata* (leaves) harvested at different altitudes on four islands of the Aeolian archipelago, in order to collect data potentially useful for interpreting the high phenotypic variability that characterizes the Aeolian populations of this species.

## 2. Results and Discussion

For the first analysis, for each location a pool of five individuals (collected at the same altitude) were extracted and analyzed by ^1^H NMR profiling. An overview of the results was acquired by performing Principal Components Analysis (PCA) using bucketed ^1^H NMR spectra as *x* variables (Figure 1A). In particular, four Principal Components (PCs) explained a maximum of 89% of the variation in the data set (given by *R*^2^*x*(cum)), while the obtained *Q*^2^(cum) was 71%, indicating very good predictability (*Q*^2^ must be equal or higher than 50%), and PC1 and PC2 explained 52.6% and 75% of the variance, respectively. 

According to this analysis, the metabolomic profile of samples collected on Stromboli differed from that of the samples collected on the other islands. In particular, samples growing on Stromboli showed a higher content of valine, proline, malic acid and asparagine. 

Plants have evolved various protective mechanisms to survive unfavorable environmental conditions, and in this regard small molecules have an enormous importance. Amino acids, for example, play an important role acting as osmolytes; some of them are involved in nitrogen assimilation or serve as precursors for the synthesis of hormones and other metabolites important for plant defense [31]. Thus, variation in amino acid concentration might reflect adaptation mechanisms to unfavorable environmental conditions.

Moreover, according to the PCA, samples harvested on Stromboli were also highly diverse. For this reason, more in-depth investigation was carried out to understand which specific metabolite variations occur in relation to altitude. Thus, the metabolome of each individual collected on Stromboli at different altitudes was analyzed, and data were treated by an Orthogonal Partial Least Squares (OPLS) model using altitude as the *y* variable (Figure 1B). 

OPLS performed well, as indicated by the coefficients: *R*^2^(cum) = 89% and *Q*^2^(cum) = 0.712%, and it was further validated by the permutation test (*R*^2^*x*(cum) = 77%; *Q*^2^(cum) = 65%) and ANOVA of the cross-validated residuals (CV-ANOVA) (*p* = 2.21 × 10^−4^, F = 9.19). This analysis confirmed that the metabolome of *D. glomerata* growing on Stromboli Island varied at different altitudes. A semi-quantitative analysis of the most varying metabolites was also performed on the basis of their ^1^H NMR diagnostic signals and is reported in the Appendix A. 

In particular, the concentration of proline, glycine betaine, sucrose, glucose and chlorogenic acid decreased at increasing altitude. The structure of the metabolite identified as chlorogenic acid by ^1^H NMR profiling was further confirmed by LC-MS analysis, showing a major peak giving an [M-H]− ion at *m*/*z* 353.23 and an [M-H]+ ion at *m*/*z* 355.33 (Appendix A).

A higher concentration of glycine betaine and proline generally reflects adaptation strategies for particular environmental conditions [32]. Glycine betaine and proline are the two major organic osmolytes accumulated in several plants in response to environmental stresses such as drought, salinity, extreme temperatures, UV radiation and heavy metals [33,34]. Moreover, the increased concentration of sucrose and glucose might reflect an increased photosynthetic activity of the plants growing at the lowest altitude or, instead, a temporary way to save energy by reducing carbohydrate consumption under drought stress, when stomatal conductance and photosynthetic activity are reduced to limit evapotranspiration. 

Conversely, increasing altitude was associated with an increment in valine, asparagine, phenylalanine and fumaric acid (Figure 2), though altitude itself could not be the main factor explaining the variability observed in the populations of Stromboli. Indeed, the volcanic activity and variations in the water availability and soil solution chemistry are the main suspects for the phenotypic and metabolomic variability of *D. glomerata* observed on the island of Stromboli. For instance, the increase in valine and phenylalanine was identified as a mechanism of tolerance to salinity in *Lygeum spartum* L. (Poaceae) [35].

The average annual rainfall in Stromboli amounts to 570 mm, with a relative humidity of 75.0% in winter and 60.8% in summer. However, the only weather station in Stromboli is located at an elevation of 4 m a.s.l., and there is a strong baric gradient between the coastline and the top of the volcano, buffering the summer drought stress with relevant condensation of atmospheric humidity [36,37]. Additionally, the hydrothermal system of Stromboli [38] causes a conspicuous degassing along the eastern slope of the volcano, where our samples were collected. The degasification induces the emission of high amounts of H_2_O and CO_2_ [39], which may locally reduce the water stress experienced by *D. glomerata* and increase its photosynthetic efficiency, thus creating the conditions for the “gigantism” displayed by the upper *D. glomerata* populations in Stromboli.

In addition, volcanic gases affect the chemical composition of soil solution and even of the rainwater [40]. The pH values of rainfall water gathered near the top craters are on average 1–2 units lower than that collected near the sea level [41]. Hence, the acidity of the soil solution may hinder potassium (K^+^) absorption, which may be the reason for the increment in valine, but this is in contrast with the observed parallel increase in asparagine, because acidic amino acids should decrease under low K^+^ concentration [42]. As a matter of fact, the Strombolian activity is characterized by high-K calc-alkaline and potassic products [43,44]; therefore, the accumulation of amino acids and amides in the tissues, due to the inhibition of protein synthesis under conditions of low K^+^ concentration, may not reflect the syndrome of the typical K deficiency.

Another possible explanation for the high levels of asparagine measured in the highest *D. glomerata* populations in Stromboli could be related to abiotic stress due to the volcanic activity, which could induce an increased biosynthesis of ethylene [45], with the toxic compound cyanide as a by-product [46]. Detoxification of cyanide involves a reaction between cyanide and cysteine to form β-cyanoalanine and subsequent conversion of β-cyanoalanine into asparagine and aspartic acid [47].

Finally, the observed increased concentration of fumaric acid could be a response of the leaf exposure to phytotoxic compounds originating from the constant discharge of acid gases and metals in the volcanic plume of Stromboli [48], although the magnitude of these interactions remains totally unknown.

Other studies have highlighted the influence exerted by altitude on the metabolome of different plants. In particular, plants of the Lamiaceae and Asteraceae family were found to be generally more enriched in phenolic compounds when growing at the highest altitude [49,50]. Generally, the metabolite variation found at increasing altitude is attributed to different factors, such as: increasing exposure to UV-B radiation, lowering of temperature, different soil features and so on [49].

However, the majority of studies are based only on spectrophotometric assays, giving unspecific chemical information, and primary metabolites, such as sugars and amino acids, are rarely investigated in relation to altitude. 

In order to acquire more information on the samples collected on Stromboli, total phenolic and flavonoid content and in vitro antioxidant activity (DPPH) of the Stromboli samples were also determined (Table 1).

Due to the strong individual variability, it was not possible to establish significative differences between samples collected at different altitudes by statistical tests. However, the samples expressing the lowest amounts of total phenolic content (1.61 ± 0.33 µg GAE/mg DW), total flavonoid content (3.76 ± 0 µg RE/mg DW) and the lowest antioxidant activity (2.76 ± 0.17 µg TE/mg DW) were collected at the highest altitude (e.g., 700 m a.s.l.). This is consistent with the decrease in chlorogenic acid, a known antioxidant phenolic compound, observed at the highest altitude. The data concerning phenolic and flavonoid content and antioxidant activity were treated by multivariate analysis (data not shown), and no correlation was found with the NMR-based profiles. 

## 3. Materials and Methods

### 3.1. Plant Sampling and Pretreatment 

*D. glomerata* was harvested at flowering stage, ca. 2 cm above soil level, in June 2019, on four islands of the Aeolian archipelago, namely: Stromboli, Vulcano, Panarea and Lipari. Sampling sites, in all islands, were set at 100, 200 and 400 m above sea level (a.s.l). Five individuals were harvested at each sampling site. In Lipari, sampling was also possible at 500 m a.s.l. and in Stromboli at 500 m and 700 m a.s.l. Sampling sites and altitude are reported in Figure 3, and coordinates are reported in Appendix A. 

Voucher specimens (Lipari BOLO0602023; Panarea BOLO0602024; Stromboli BOLO0602025; Vulcano BOLO0602022) were deposited at Department of Pharmacy and Biotechnology, University of Bologna (via Irnerio 42, Bologna, Italy). After sampling, the fresh plant material was dried in ventilated stove at 40 °C to constant weight and ground in an electric grinder.

### 3.2. Chemicals

Deuterium oxide (D_2_O, 99.90% D) and CD_3_OD (99.80% D) were purchased from Eurisotop (Cambridge Isotope Laboratories, Inc, France). L-proline standard was purchased from ICN Biomedicals. Standard 3-(trimethylsilyl)-propionic-2,2,3,3-*d*_4_ acid sodium salt (TMSP), sodium phosphate dibasic anhydrous, sodium phosphate monobasic anhydrous, L-phenylalanine standard and all the other solvents and chemicals were purchased from Sigma-Aldrich Co. (St. Louis, MO, USA). 

### 3.3. Extract Preparation for NMR Analysis

Thirty milligrams of dried and powdered leaves were extracted using 1 mL of a blend (50:50) CD_3_OD/D_2_O (containing 0.1 M phosphate buffer and 0.01% of TMSP standard). Samples were sonicated for 20 min and subsequently centrifuged for 10 min at 17,000× *g*; the supernatant (700 µL) was then separated from the pellet and transferred into NMR tubes.

For the PCA, each sample represents a pool of the five individuals collected at the different altitude. Conversely, for the OPLS, each sample is an individual harvested in Stromboli; thus, for each altitude, five samples were prepared. 

The choice of the extraction solvents was based on metabolomics works [21,51], where MeOH/H_2_O (1:1) turned out to be the best choice for first-line extraction of generic plant material, leading the obtainment of a broad spectrum of compounds.

### 3.4. Extract Preparation for Spectrophotometric Assays

*D. glomerata* samples collected on Stromboli were extracted to determine total phenolic content, total flavonoid content and in vitro antioxidant activity by means of spectrophotometric assays. 

The extraction procedure was analogous to the one used for NMR analysis. Namely, 30 mg of dried and powdered leaves underwent ultrasound-assisted extraction (20 min) using 1 mL of a blend (50:50) methanol–water. After 10 min of centrifugation at 17,000× *g*, the supernatant was separated from the pellet and subjected to the spectrophotometric assays. 

### 3.5. NMR Spectra Measurement

NMR spectra were recorded at 25 °C on a Varian Inova instrument (equipped with a reverse triple-resonance probe). For ^1^H NMR profiling, the instrument operated at ^1^H NMR frequency of 600.13 MHz, and H_2_O-*d*_2_ was used as internal lock. Each ^1^H NMR spectrum consisted of 256 scans (corresponding to 16 min) with a relaxation delay (RD) of 2 s, acquisition time 0.707 s and spectral width of 9595.8 Hz (corresponding to δ 16.0). A presaturation sequence (PRESAT) was used to suppress the residual water signal at δ 4.83 (power = −6 dB, presaturation delay 2 s). Each NMR spectrum was recorded once.

### 3.6. NMR Processing and Multivariate Data Treatment

Free Induction Decays (FIDs) were Fourier transformed, and the resulting spectra were phased, baseline-corrected and calibrated to TMSP at δ 0.0, which was also used as a standard for semi-quantitative analysis (Appendix A). Spectral intensities were reduced to integrated regions of equal width (δ 0.04) corresponding to the region from δ 0.0 to 10.0, with scaling on total area using the NMR Mestrenova 12 software (Mestrelab Research, Santiago de Compostela, Spain). The analysis of ^1^H NMR profiles was performed based on comparison with the literature and an in-house library [29,52]. In particular, proline and phenylalanine standard were compared to the spectra of the samples in order to support their tentative identification (Appendix A). L-phenylalanine and L-proline were dissolved in the same blend of solvents used for samples, at concentration of 2 mg/mL.

The regions of δ 5–4.5 and 3.34–3.30 were excluded from the analysis of the aqueous samples because of the residual solvent signals. For multivariate analysis, the models (PCA and OPLS) were developed using SIMCA-P software (v. 16.0, Umetrics, Umeå, Sweden). Data were subjected to Pareto scaling. The supervised model OPLS was evaluated by the goodness of fit (*R*^2^*y*(cum)) and goodness of prediction (*Q*^2^(cum)), together with the parameters given by the permutation test (performed using 100 permutation) and CV-ANOVA [53]. 

### 3.7. UHPLC−MS Analysis

UHPLC−MS analysis was run on a Waters ACQUITY ARC UHPLC/MS system consisting of a QDa mass spectrometer equipped with an electrospray ionization interface and a 2489 UV/Vis detector. The detected wavelengths (λ) were 254 nm and 365 nm. 

The analyses were performed on an XBridge BEH C18 column (10 × 2.1 mm i.d., particle size 2.5 μm) with a XBridge BEH C18 VanGuard Cartridge precolumn (5 mm × 2.1 mm i.d., particle size 1.8 μm). 

The mobile phases were H_2_O (0.1% formic acid) (A) and MeCN (0.1% formic acid) (B). Electrospray ionization in positive and negative mode was applied in the mass scan range of 50−1200 Da. 

The hydroalcoholic extract of *D. glomerata* growing on Stromboli at 100 m a.s.l. was diluted to 200 µg/mL, and a volume of 3 µL was injected. The extract was eluted with the following method: 5% B for one minute followed by a gradient reaching 25% B in 2 min; 25% B was kept for 1 min; then the gradient reached 70% B in 2 min; 70% B was kept for 1 min; and then the gradient reached 5% B again in 2 min and was kept at this concentration for 1 min. Flow rate was 0.8 mL/min. 

This analysis allowed chlorogenic acid to be characterized (Appendix A). 

### 3.8. Total Phenolic and Total Flavonoid Content

The total phenolic and total flavonoid content of *D. glomerata* growing on Stromboli were assessed by means of spectrophotometric methods using the spectrophotometer Victor™ X3 PerkinElmer (Waltham, MA, USA) as described by Chiocchio et al. [54] with slight modifications. 

The crude extracts were diluted in methanol (1:8) and tested in duplicate for each assay. Gallic acid and rutin were used to build the calibration curves, which were used to calculate by interpolation the total phenolic content and total flavonoid content, respectively. Thus, total phenolic content was expressed as µg of gallic acid equivalent (GAE) per mg of dried plant material (DW, e.g., dried weight), and total flavonoid content was expressed as µg rutin equivalent (RE) per mg of dried plant material.

### 3.9. Antioxidant Activity

The in vitro antioxidant activity of *D. glomerata* growing on Stromboli was determined by DPPH free radical scavenging assay [55]. Crude extracts were diluted in methanol in order to test different concentrations (ranging from 7.8 to 250 μL/mL, stock solutions). A total of 50 μL of sample solutions was added to 700 μL of a DPPH methanol solution (50 mM). After 20 min at room temperature, absorbances (Abs) were measured at 517 nm, and the percentage of antioxidant activity was calculated using the following formula:Antioxidant activity % = [(control Abs − sample Abs)/control Abs] × 100

Methanol was used as negative control, and Trolox (Tr) at different concentrations (ranging from 50 to 300 μM) was used as a positive control, and Tr IC_50_ (14 μM) was used for the calculation of Trolox equivalents. Total antioxidant activity was expressed as µg of Trolox equivalent (TE) per mg of dried plant material.

## 4. Conclusions

In conclusion, *D. glomerata* metabolome was strongly impacted by environmental conditions. In particular, samples collected on Stromboli had intensified accumulation of specific metabolites according to the collection site. Moreover, the metabolomic profiles of *D. glomerata* samples growing on Stromboli were different from those of the samples growing on the other islands. Measurements of deposition chemistry, meteorology and soil solution chemistry, as well as assessment of the seasonal variability in the metabolomic profile, would be necessary to confirm the hypotheses made in this preliminary study. Moreover, the local environmental drivers related to the altitudinal gradient and to the volcanic activity might not be the only causes for the differences in the metabolomic profile of samples collected on Stromboli compared to the samples collected on the other islands. According to Carpaneto (1985) [56], the *Coleoptera Scarabeoidea* of Stromboli have a greater affinity with the fauna of Calabria (Southern Italy) than any other Aeolian island. Further studies, conducted through phylogenetic analyses, would be needed to quantify the relative affinity of the populations of *Dactylis glomerata* between the different islands and in relation to the two nearest mainlands, i.e., Sicily and Calabria.

## Figures and Tables

**Figure 1 metabolites-12-00533-f001:**
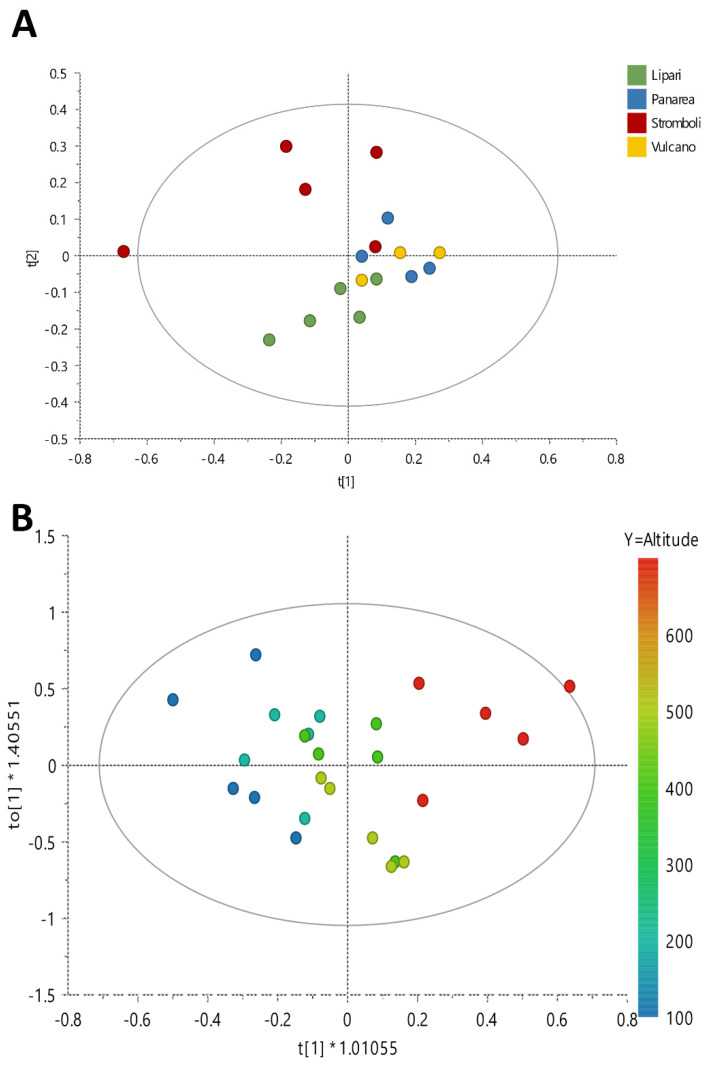
(**A**) PCA Score Scatter Plot and (**B**) OPLS Score Scatter Plot (*y* = altitude).

**Figure 2 metabolites-12-00533-f002:**
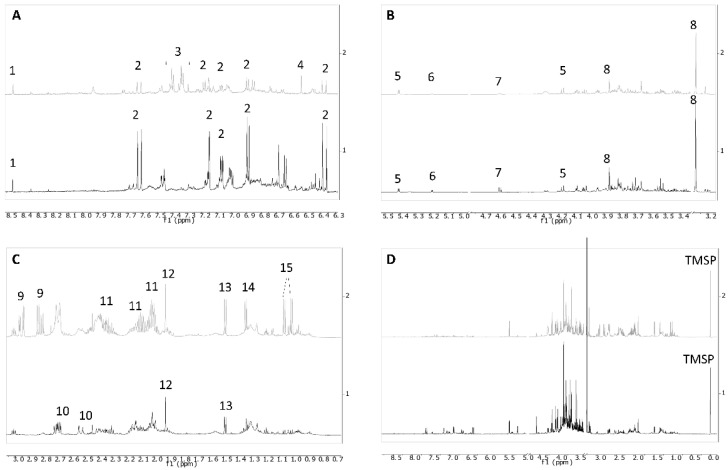
^1^H NMR profiles of representative samples harvested at 700 m a.s.l. (gray trace) and 100 m a.s.l. (black trace) on Stromboli. (**A**) Extended spectral region from δ 6.3 to 9; (**B**) region from δ 3.2 to 5.5; (**C**) region from δ 0.4 to 3.1; (**D**) full spectrum. Regions 4.78–4.98 and 3.30–3.35 were cut because of residual solvent signals. 1 = formic acid, 2 = chlorogenic acid, 3 = phenylalanine, 4 = fumaric acid, 5 = sucrose, 6 = α-glucose, 7 = β-glucose, 8 = glycine betaine, 9 = asparagine, 10 = malic acid, 11 = proline, 12 = acetic acid, 13 = alanine, 14 = threonine, 15 = valine.

**Figure 3 metabolites-12-00533-f003:**
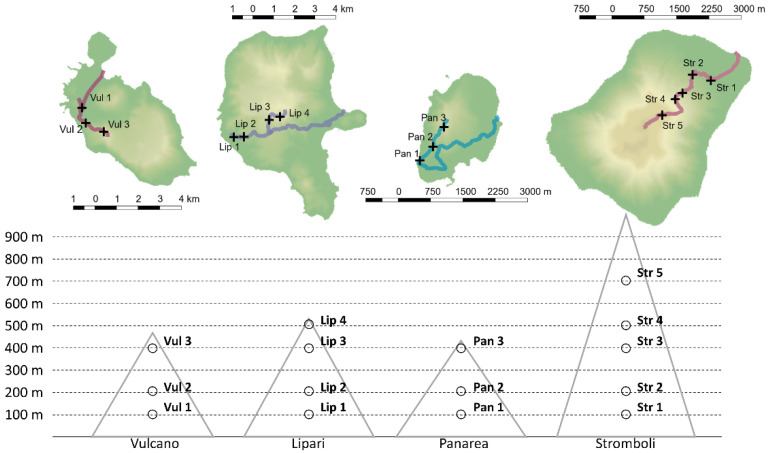
Collection sites. Map of the islands created in QGIS 2.18.20 (QGIS Development Team, 2016) using the DEM file provided by ISPRA and the shape file provided by ISTAT.

**Table 1 metabolites-12-00533-t001:** Total phenolic content, total flavonoid content and antioxidant activity of samples collected on Stromboli Island and different altitudes. Values are expressed as mean ± standard deviation.

	Total Phenolic Content (µg GAE/mg DW)	Total Flavonoid Content (µg RE/mg DW)	Antioxidant Activity (µg TE/mg DW)
**100 m a.s.l.**	7.25 ± 0.32	8.34 ± 0.18	17.05 ± 1.66
	5.82 ± 0.1	6.13 ± 0.42	10.76 ± 0.72
	11.6 ± 0.4	15.8 ± 0.2	12.25 ± 0.25
	15.77 ± 0.34	17 ± 1.13	21.79 ± 2.77
	20.92 ± 0.14	27.48 ± 0.63	17.95 ± 0.68
**200 m a.s.l.**	15.66 ± 0.63	19.09 ± 0.43	28.64 ± 3.14
	11.47 ± 0.23	12.37 ± 0.29	23.34 ± 1.55
	19.38 ± 0.4	33.72 ± 0.4	27.23 ± 0.64
	10.42 ± 0.03	14.11 ± 0.27	21.57 ± 1.3
	12.3 ± 0.09	17.39 ± 0.38	31.81 ± 0.41
**400 m a.s.l.**	7.98 ± 0.11	12.08 ± 0.27	16.12 ± 2.94
	29.33 ± 1.27	15.09 ± 0.76	19.38 ± 1.7
	22.27 ± 0.56	10.26 ± 0.15	9.52 ± 1.14
	9.09 ± 0.19	10.95 ± 0.2	13.22 ± 1.03
	10.75 ± 0.12	15.95 ± 0.02	20.71 ± 1.68
**500 m a.s.l.**	9.57 ± 0.03	11.47 ± 0.05	7.21 ± 1.06
	9.7 ± 0.19	13.03 ± 0.13	13.83 ± 0.09
	8.67 ± 0.18	12.4 ± 0.59	7.66 ± 1.16
	6.12 ± 0.95	6.82 ± 0.45	9.28 ± 0.21
	7.24 ± 0.63	9.16 ± 0.05	9.3 ± 0.55
**700 m a.s.l.**	1.61 ± 0.33	7.63 ± 0.05	8.31 ± 1.21
	9.67 ± 1.61	8.46 ± 1.11	12.04 ± 2.76
	6.27 ± 1.4	7.62 ± 0.29	8.22 ± 0.2
	4.97 ± 0.29	7.25 ± 0.33	6.37 ± 0.63
	3.65 ± 0.85	3.76 ± 0	2.76 ± 0.17

## Data Availability

The data presented in this study are available in the main article and the Appendix A.

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
