# Peer review of "Metabolomic Study of Dactylis glomerata Growing on Aeolian Archipelago (Italy)"

_metabolites, 2022, doi:10.3390/metabo12060533_

Round 1

Reviewer 1 Report

The paper has been improved in accordance with my suggestions. In my opinion, the paper can be accepted to publication.

Author Response

We would like to thank reviewer 1 for the contribution to improve this manuscript.

Reviewer 2 Report

The revised revision reported in this manuscript represents a further improvement in the ongoing efforts of the study to further characterize the antioxidant and LC/MS analyses and these analyses results, which like most of revisions got great results for submission. However, new results presented in Line 269-280 should be more precisely to report the analysis method and the result. For example, the sentences in the Line 269-283 should be combined together into one paragraph, and the flow rate should be not in the high 0.8 mL/min owing to the column (10 × 2.1 mm i.d., particle size 2.5 μm) used in the study. The type writing of Figure S1 should be revised to Figure S3. In addition, the data of Figure S3 shown in the supplementary materials should be revised again for the correct presentation according to the authors’ described UHPLC-MS analysis, which means that the data of total ion chromatogram of LC-MS analysis and the MS spectra of indicated compound chlorogenic acid should be shown together in the Figure, but not only the MS spectra shown in the Figure S3.
The second question is that the discussion of antioxidant activity on the selected plant should be updated since you have already described in the Materials and Methods and cited the reference 13 in the manuscript.

Generally speaking the manuscript is well-written and clear revised, however it could benefit from revision again as indicated in the detailed comments as the above. After that the manuscript would be considered as suitable for the publication of the journal.

Author Response

We would like to thank reviewer 2 for the contribution to improve this manuscript.

Please find our response point-by-point below:

1) The reviewer states that the sentences in the Line 269-283 should be combined together into one paragraph, however, in these lines are reported UPLC-MS method and phenolic and flavonoid content analytical method. We believe these two are too different analytical methods, in fact, they are generally given separately, so we would prefer to keep them in different paragraphs.

2) The flow rate used is 0.8 mL/min, the method was used only to confirm the structure of chlorogenic acid, and this flow rate resulted good for our objective.

3) We have corrected the numeration of supporting material figures (line 256).

4) We have now included in Fig. S3 also the chromatogram in addition to the MS analysis.

5) As suggested during the first-round revision, we have analysed the antioxidant activity of the samples. We have now reported that the data were also treated by multivariate analysis and no correlation with the NMR profiling was found (Line 192-194). Having not found any significative difference among the antioxidant activity of the samples collected at the different altitude, we prefer to leave the discussion simple as it is. Morover, reference 13 reported the antioxidant activity of some pure compounds isolated from Dactylis glomerata, thus it is not possible to copare our data (that are performed on crude extracts) with that of reference 13. We did a further bibliographic research and we found articles in which it is assay the antioxidant activity of Dactylis growing under special conditions, like exposure to heavy metals and extreme CO2 concentration, but the methods used and the context are too different from our work, we are afraid that any further discussion could only weigh down the manuscript.

Reviewer 3 Report

Findings obtained in this paper revealed that environmental conditions affects strongly the content of the secondary metabolites in Dactylis glomerata growing on four islands of Aeolian archipelago (Italy). These conclusions were developed using multivariate approach (PCA and OPLS). However, multivariate calculations were performed using only NMR data.

Authors should explain, why the data on total phenolic content, total flavonoid content, antioxidant activity (DPPH test), and chlorogenic acid (UHPLC-MS) which were determined in the extracts prepared from Dactylis glomerata, were not used for multivariate calculations.

Moreover, the authors showed that the first four PCs explained 89% of the variance in the data set. As Figure 1A is constructed using only two PCs, information is missing, how many percent of total variance is explained by these two PCs.

Author Response

We would like to thank reviewer 3 for the contribution given to improve this manuscript.

Please find our response point-by-point below:

1) We have now explained in the text (lines 191-193) that data concerning phenolic and flavonoid content and antioxidant activity were treated by multivariate analysis but no correlation was found with the NMR-based profiles. This was quite obvious since no significant statistical differences were found among phenolic and flavonoid content and antioxidant activity of the samples collected at the different altitudes (as it is reported in the text).

2) Regarding the UPLC-MS, it was used only to confirm the presence of a metabolite, namely chlorogenic acid, whose structure assignation was not fully clear from the 1H NMR profile. This analysis was performed only on one sample, as it is stated in the manuscript, then it was not subjected to multivariate data analysis.

3)  We have now reported how many percent of total variance is explained by these two PCs used in Fig. 1 (lines 79-80)

This manuscript is a resubmission of an earlier submission. The following is a list of the peer review reports and author responses from that submission.

Round 1

Reviewer 1 Report

The paper entitled ‘Metabolomic study of Dactylis glomerata growing on Aeolian archipelago (Italy)’ aimed at comparing the metabolomic profiles of D. glomerata (leaves) harvested at different altitudes on four islands of the Aeolian archipelago. On the whole, the subject is interesting and could be helpful for scientists and other readers nevertheless the paper is not good prepared. The abstract and Introduction are informative, materials are described in detail, but the methodology should be enriched with additional studies. The NMR analysis is insufficient and the paper should include studies such as HPLC-MS as well as assays towards biological activities (i.e. antioxidant, total amount of polyphenols, terpenes, etc.) which would allow to compare the various extracts. Additionally, the results and discussion section should be presented in more detail. Discussion is insufficient. References include several publications from several years ago that should be replaced with more up-to-date literature.

Reviewer 2 Report

Metabolomic study of Dactylis glomerata growing on Aeolian archipelago (Italy)

The study aimed to evaluate the differences of plant metabolites of Dactylis glomerata among the four islands of the Aeolian archipelago by using 1H NMR.
Although the studies have indicated the differences of metabolites among the samples, the following issues need to be taken into account to improve the quality.
1. The results of 1H NMR analysis in the study are limited to the small molecules (M.W. < 200). Information on what the other metabolites, i.e. polyphenols, should be stated in the study. The reference Journal of Agricultural and Food Chemistry, 2013, 62(2), 468–475 might be one of the sources.  

  1. The authors should give the appropriate references to describe the limitation of 1H NMR analysis on plant metabolites.
  2. The authors need to introduce something about the composition of metabolites of Dactylis glomerata.
  3. It appears that the extraction solvent of using 1 mL of a bland (50:50) CD3OD/D2O in the study might obtain the extracts with more hydrophilic compounds. The solvent choice for the sample extraction in the study should be more integrally. The authors need to discuss something about the solvent decision in the study.

Overall, I would encourage the authors to address the aforementioned concerns that were identified above and resubmit it for improving the quality of the manuscript. .

Reviewer 3 Report

The manuscript entitled “Metabolomic study of Dactylis glomerata growing on Aeolian archipelago (Italy)” seems to be interesting, however, nothing new from the metabolomic profiles point of view. The data on bioactive ingredients are very little. In my opinion, the quality of this manuscript has low priority to be published in Metabolites. The reasons for this are shown below.

First of all, the paper is too general and does not include any details about metabolites quantitation. Hence, the detailed information about tentative concentrations of metabolites in Dactylis glomerata extracts should be inserted into the manuscript.

Secondly, the authors should comment what is the reason that they used several different multivariate statistical tools, that is – principal component analysis (PCA), OPLS, OPLS-DA, PLS-DA and CV-ANOVA. Additionally, the acronyms of OPLS, OPLS-DA, PLS-DA and CV-ANOVA are not explained in the manuscript. According to the general rule, abbreviations and acronyms should be defined for the first time when they are used. Only the results for PCA and OPLS are shown in the manuscript. What about the findings obtained using remaining multivariate tools? How do these multivariate tools correlate with each other?

Please explain in more detail, what multivariate measurement data were used to calculate PCA, OPLS, OPLS-DA, PLS-DA and CV-ANOVA? PCA was performed using covariance or correlation matrix? Does the data used for PCA were pre-processed before calculations? What about validation of PCA method?

Reviewer 4 Report

The subject of the manuscript is interesting, although the work would be more interesting and reliable if the research was carried out on a research material were more diverse (for at least two seasons). Important information would also be the analysis of the climatic conditions of the region and the analysis of the soil at the sampling site.

In the introduction of the work, attention should be paid to the innovative nature of the research presented in the article. What is the innovative aspect of the research described? What information does this research bring to the field of science about the metabolomics? Did other studies also show a relationship between the concentration of various compounds in other plants and altitude? This aspect should be discussed in the introduction to the research.

The analytical methods used in the article are standard, the sample preparation, the description of the analytical equipment and the purity of the reagents are described in detail. The description of the experiential design is insufficient. How many times have the NMR spectra been performed? We know that five individuals were collected at each sampling site. After drying and grinding, only thirty mg of this material was analyzed? Were the analyzes performed in duplicate?

I recommend developing a discussion on the production of various compounds by plants in order to protect them under unfavorable environmental conditions (L72). The studies showed significant differences in the content of valine, proline, malic acid and aspartate in particular samples growing on Stromboli. What function do these compounds play in the plant and its defense mechanism? There are earlier publications and therefore I propose to add information on this subject.

The text of the article is written comprehensively, while Figures 2 and 3 raise objections to legibility. I recommend that you improve the legibility of these figures.